# Strengthening the Voices of Hispanic/*Latine* Immigrants Managing Chronic Disease: A Mixed Methods Approach to Understanding Perspectives of Health

**DOI:** 10.3390/healthcare12151519

**Published:** 2024-07-31

**Authors:** Kathy Zamarripa, Ambria Crusan, Kerrie Roozen, Clara Godoy-Henderson, Angela Evans

**Affiliations:** 1Department of Biology, Aging & Longevity, School of Arts, Humanities, and Sciences, St. Catherine University, St. Paul, MN 55105, USA; kzamarripa540@stkate.edu; 2Department of Nutrition and Dietetics, St. Catherine University, St. Paul, MN 55105, USA; 3Department of Health Services Research, Policy and Administration, School of Public Health, University of Minnesota, Minneapolis, MN 55455, USA

**Keywords:** health perception, intersectionality of wellness, immigrant health, dimensions of health, culturally competent healthcare, Hispanic, Latino, Latina, Latine

## Abstract

Individuals who migrate from their home country face a variety of challenges while adapting to the culture in the United States. Immigrant communities are at a significantly higher risk for poor health outcomes; therefore, assessing healthcare treatment for diverse and resilient immigrant populations, including Hispanic/*Latine* communities, is crucial to preserving their health, culture, and spirit. A paucity of literature exists surrounding perceptions of well-being in immigrant, Hispanic/*Latine* adults managing chronic diseases. Past studies have shown a discrepancy between providers’ and patients’ perceptions of healthcare options for overall well-being. We aim to share varying perspectives found within our work geared towards improving the quality of life for Hispanic/*Latine* immigrants managing chronic disease, especially type 2 diabetes mellitus and hypertension. The primary objective of this article is to strengthen the understanding of intersections between social, physical, financial, and spiritual health within an (im)migrant Hispanic/*Latine* community using semi-structured ethnographic interviews. These interviews have highlighted community resilience, demonstrating that individuals can adapt to major life transitions while maintaining balance across dimensions of health. This knowledge could be implemented by actively listening to patient concerns regarding their health dimensions to improve individualized and patient-centric care.

## 1. Introduction

Assessing healthcare treatment for diverse and resilient immigrant populations, including Hispanic/*Latine* communities, is crucial to preserving the unique health, culture, and spirit of the United States’ diversity [1,2]. Systemic racism has historically hindered how migrants perceive health and healthcare because of systems of oppression in the United States (US). Individuals who migrate from their home country may face increased barriers related to social determinants of health, including food access and affordability, transportation, literacy, healthcare access and education, and social support [1]. Moreover, the healthy immigrant effect, a connection between the length of residency in the US, access to quality healthcare, and acculturation to a Westernized diet, shows an increased risk of health inequities for people managing chronic diseases within migrant populations [3,4,5].

Immigration has been shown to alter physical health, and additional studies show immigration impacts additional health dimensions. While the literature surrounding the perception of physical and mental well-being in Hispanic/*Latine* adults is scarce, some research suggests this population does not seek Western healthcare services because of traumatic or stressful experiences [2]. Additionally, providers’ low cultural awareness of health perceptions can result in poor health outcomes for immigrant populations [1]. This concept is reinforced by Moyce et al., who demonstrate that lifestyle alterations and acculturation stressors associated with immigration can have implications on healthcare-seeking behaviors [2]. Therefore, it is important to recognize the growing movement for healthcare practitioners to assess multiple health dimensions, encompassing a more holistic approach to chronic disease management.

One approach to wellness considers multiple dimensions of health to represent the intersectionality of health and promote individualized (self-)care and, thus, health equity. Five of the dimensions of health, (1) social, (2) physical, (3) financial, (4) spiritual, and (5) mental, adapted from Stowen 2017, are important factors pertaining to an individual’s life experience, health literacy, and overall quality of life (Figure 1) [6]. Consideration should be given to the health dimensions when providing healthcare as access to food and employment (financial health), social support and engagement (social/spiritual health), and health literacy (mental/physical health) greatly impact an individual’s wellness. Providing optimal care without bias and stigma while understanding and recognizing how external factors, like immigration, impact the dimensions of wellness can provide a framework to improve medical mistrust.

A biocultural approach to reviewing mental illness experience and expression literature by Shattuck emphasizes the impacts of immigration within the intersectionality of genetics, culture, environment, and other social determinants of health [7]. Immigrant populations come from a variety of different backgrounds and hold variable cultural values related to health. More specifically, those identifying as Hispanic/*Latine* may place increased value on family or religion and seek to strengthen familial bonds for the promotion of social or spiritual health [2]. Additionally, family and social networks provide healthcare support. For example, Hispanic/*Latine* individuals who migrate to the US are often supported by their community to follow their traditional dietary patterns and maintain healthful cultural practices [4]. A more inclusive approach to working with Hispanic/*Latine* immigrant populations recognizes the collectivist culture promoting social health values and prioritizes connections with their family and immediate community [2,8]. Recognizing that these dimensions of health overlap and impact one another could aid in the treatment of chronic and complex diseases.

Understanding the intersectionalities of health dimensions for Hispanic/*Latine* American immigrant populations remains underresearched. Therefore, this study aims to: (a) explore the self-perceptions of five health dimensions in a sample of Hispanic/*Latine* individuals managing a chronic disease via survey, and (b) understand the intersections of health as perceived by Hispanic/*Latine* individuals experiencing major life transitions, such as immigration, while addressing the immigrant experience in relation to chronic disease management via ethnographic interviews.

## 2. Materials and Methods

This study is a 2-phased subanalysis of a larger study utilizing community-based participatory research methods to understand culturally appropriate food choices. Understanding how the participants perceived health was an important component to building our medically tailored grocery intervention. Therefore, the survey responses from 70 participants (7 Hispanic/Latine immigrant healthcare providers, 63 Hispanic/Latine immigrant patients) came from the first phase utilizing a quantitative community survey and the qualitative data were collected in a second phase via ethnographic interviews with 14 participants [9].

### 2.1. Quantitative Community Survey (Phase 1)

#### 2.1.1. Participants

Utilizing community-based participatory research practices to understand perceptions of health, the quantitative portion of the study began in June of 2022. Inclusion criteria for survey participation were: 18+ years of age, a patient or provider (such as nurse, medical interpreter, or community health worker) at a local community health clinic, self-identifying as Hispanic/*Latine*, and an immigrant to the US. Participants were able to select their preferred language (Spanish or English) to complete the survey. To be patients at the community health clinic in which we surveyed study participants, the individuals had to meet the following requirements: (1) being uninsured or underinsured, (2) living within 200% of the Federal Poverty Guidelines, and (3) maintaining compliance to scheduled appointments. Patients are often recruited to the clinic through their country’s consulate or social networks.

#### 2.1.2. Data Collection

A 21-question survey was administered during 6 in-person clinic dates between June and October of 2022 via paper survey to 70 patients and providers receiving care for chronic diseases at a community health clinic in Minnesota. Participants with an appointment at the community health clinic or providers working at the clinic on the random clinic days were given the option to participate if they met the inclusion criteria. Since the study was exploratory, there were no intended outcomes on sample size number, but achieved a final survey number after the summary of the 6 clinics in which the surveys were collected. The results from the survey were individually recorded into the electronic management platform, Research Electronic Data Capture Version 14.0.1 (REDCap) [10,11]. This data management resource is hosted by St. Catherine University as a secure web-based software platform designed to support data capture for research studies, providing: (1) an intuitive interface for validated data capture; (2) audit trails for tracking data manipulation and export procedures; (3) automated export procedures for seamless data downloads to common statistical packages; and (4) procedures for data integration and interoperability with external sources.

#### 2.1.3. Measures

Quantitative data regarding social determinants of health and health perceptions were assessed via demographic questions from the PhenX toolkit [12], Patient-Reported Outcomes Measurement Information System (PROMIS) Scale V 1.2—Global Health instrument [13], and the United States Department of Agriculture (USDA) Six-Item Short Form of the Food Security Survey [14]. An exploratory analysis of the survey data was conducted to increase the understanding of participants’ self-perceptions related to overall health and five health dimensions adapted from Stowen [6]. Demographic questions were obtained from the PhenX toolkit to collect information on age, biological sex, gender identity, sexual identity, race, ethnicity, birthplace, education, and employment status [12].

The PROMIS Scale V 1.2, Global Health instrument, a validated measure assessing physical, mental, and social health in adults living with chronic conditions, was used to assess overall perceptions of health. The question Global 01, “In general, would you say your health is” was rated on a scale of 1–5, 1 being poor and 5 being excellent. Physical health was measured using the PROMIS Global Physical Health T-Score, which was calculated by scoring the PROMIS Scale V 1.2, Global Health instrument questions Global 02, 04, 05, and 10r and scored via the Global Health Scoring Manual. Similarly, mental health was measured using the PROMIS Global Mental Health T-Score, which was calculated by scoring the PROMIS Scale V 1.2, Global Health instrument questions Global 03, 06, 08r, and 7r and scored via the Global Health Scoring Manual. Social health was assessed via an average of two questions, Global 05, “In general, how would you rate your satisfaction with your social activities and relationships?” and Global 09r, “In general, please rate how well you carry out your usual social activities and roles,” which were both rated on a scale of 1–5, 1 being poor and 5 being excellent [13].

The USDA Six-Item Short Form of the Food Security Survey, which is an effective tool to determine food security status with high specificity and sensitivity with minimal bias [14], was used to determine food security status. The data were categorized through a tiered food security analysis of “high or marginal security”, “low security”, or “very low security” using the reported analysis measures provided by the survey [15].

#### 2.1.4. Quantitative Analysis

For the analysis of the quantitative survey results, JASP statistical software was utilized (JASP Version 1.17.2, Computer software). Descriptive statistics were used to understand the means and standard deviation of responses to each of the measures related to the health dimensions (overall health, Mental Health T-Score, Physical Health T-Score, and social health rating). *T*-tests were used to assess significant differences between measures within demographic subgroups, with significance at *p* < 0.05. Pearson correlations were used to understand the strength of the relationships between measures related to the health dimensions and (1) sex, (2) age, and (3) food security status.

### 2.2. Qualitative Interviews (Phase 2)

#### 2.2.1. Participants

The qualitative portion of the study included a subset of the quantitative survey respondents (patients only, not providers) who agreed to a follow-up for a 1-on-1 ethnographic interview. A 1 h, semi-structured interview was conducted with 14 patients between October 2022 and January 2023. Participants were referred to the study coordinator if they indicated interest in a more in-depth interview to explain their perceptions of health. Initially, 23 patients who indicated they were managing a chronic disease in the qualitative survey and selected the option to be contacted for follow-up studies were recruited to participate in the interview; however, 8 participants canceled or did not show and 1 participant did not complete the interview due to time constraints. Participants were screened to ensure they met the inclusion criteria of: 18+ years of age, a patient at the community health clinic, self-identifying as Hispanic/*Latine*, and managing a chronic disease.

#### 2.2.2. Data Collection

This study utilized the survey data to develop questions for formative intervention to better understand perceptions of health and the intersectionality of health dimensions via 14 semi-structured, ethnographic interviews. An ethnographic interview method was chosen as an effective way to facilitate cross-cultural understanding [16]. The interview began with an overview of the study, introductions to the researchers and their role in the community health clinic and university, the purpose of the interview, the proposed length of the interview, and the right to refuse to answer any questions for any reason. Participants were then asked to organize dimensions of health by level of importance (see Appendix A for interview questions). Figure 1 was not shown to participants before interviews; the definitions of each of the referenced health dimensions were open to their interpretation.

Interviews were conducted by two members of the research team (A.C., K.Z., K.R., C.G.-H., and A.R.). To ensure the interview was in the participant’s preferred language (Spanish or English), one research team interpreter (A.R. or C.G.-H.) and one interview facilitator were present. The interviews that were conducted in Spanish were translated by our interpreters, who are native Spanish speakers (A.R. and C.G.-H.), and professionally transcribed prior to data analysis. Due to varying schedules, barriers to transportation, and time, all participants were given the option to complete the interview in person or virtually (in person *n* = 4, virtual *n* = 10). All interviews were recorded with the permission of the participant. The Standards for Reporting Qualitative Research Checklist was used to ensure that the study methods were reported accurately [17].

#### 2.2.3. Analysis

Interviews were transcribed and themes were identified regarding intersectionalities of health dimensions using framework analyses. More details on our data analysis methods are cited in our previous work [9]. Saturation on themes was reached after 10 interviews.

For financial health, a financial barrier score was created using three measures. Alternative measures were utilized, as all patients eligible to receive care at the community health clinic must live within 200% of the Federal Poverty Guidelines, making financial data irrelevant. One measure utilized was the USDA Six-Item Short Form of the Food Security Survey [14] with “yes” or “no” regarding food insecurity. The participant was assigned “1” if they were food insecure. The second consideration was employment status. This was used to gauge income and participants categorized as “employed” (earning income from full or part-time work) were coded as 0, and “unemployed/retired” (including those unemployed, retired, keeping house, or temporarily laid off/on leave) were coded as 1. The third category of financial barrier scoring included a verbal indication of money being a health barrier during ethnographic interviews. A total score of “0” was coded if no financial barriers were present, and a score of “3” indicated all financial barriers were present, with a higher overall score indicating increased likelihood of financial barriers experienced by the participant.

The study was conducted in accordance with the Declaration of Helsinki and approved by the Institutional Review Board of St. Catherine University (protocol code 1744, with date of approval on 9 May 2022). Written or verbal informed consent was provided by all participants prior to engaging in the survey and/or the interview. The research team emphasized that there was no influence on the participants’ right to receive healthcare based on participation in the study. Due to the principal investigator’s role in providing nutrition assessments at the community health clinic, the research team ensured that the principal investigator did not provide care to any of the participants prior to the study.

## 3. Results

### 3.1. Quantitative Survey

#### 3.1.1. Participant Demographics

Among the 70 participants who completed the 21-question survey, the average age was 47.8 ± 13.0 years. All participants were of Hispanic/*Latine* ethnicity (countries of origin: Mexico, Puerto Rico, Guatemala, Ecuador, Cuba, Honduras, El Salvador, and Venezuela), with the highest proportion of participants from Mexico (71.4%). The assessment of the tiered outcome for food security showed 47.1% of the sample indicated food insecurity as low security (*n* = 31) or very low security (*n* = 2) while 50.0% indicated high or marginal food security (*n* = 35). Two participants declined to complete the food insecurity screener. Significant differences were found in overall health rating, mean Mental Health T-Score, and mean Physical Health T-Score between participants with high food security and those with low/very low food security. Participants’ mean Mental Health T-Score, mean Physical Health T-Score, and social health score are reported in Table 1 by sex, employment status, and food security status. The sample’s average Global Mental Health T-Score (48.0 ± 3.7) and Physical Health T-Score (46.6 ± 4.5) were both slightly lower than the US reference population T-Score of 50 [18,19].

#### 3.1.2. Correlations

We found no significant correlations between age or sex and the Mental Health T-Score or the Physical Health T-Score. There were also no significant correlations between age, sex, or food insecurity and the social health rating. However, moderate correlations were present between food insecurity and Mental Health T-Score (r = −0.30, *p* < 0.05) and Physical Health T-Score (r = −0.30, *p* < 0.05), indicating higher food insecurity is related to decreased physical and mental health perceptions.

#### 3.1.3. Financial Barrier Scoring

As referenced in Table 2, 35.7% of participants (*n* = 5) reported a total score of 3 for their financial barrier scoring, meaning they had three factors (food insecurity, employment status, and reported financial barrier) influencing their financial health. Additionally, 85.7% of participants (*n* = 12) reported at least one factor contributing to their financial health.

### 3.2. Qualitative Interviews (Phase 2)

#### 3.2.1. Participant Demographics

Among the 14 participants (11 women, 3 men) that completed the 1-on-1 interviews, the average age was 56.5 ± 7.7 years, ranging from 42–68 years. All of the participants were of Hispanic/*Latine* ethnicity (country of origin: Mexico, Puerto Rico, Guatemala, Ecuador, and Venezuela), with self-reported races being black (*n* = 1), white (*n* = 11), and Native American Aztec (*n* = 2). Chronic diseases managed by participants were hypertension (*n* = 9), dyslipidemia (*n* = 4), pre-diabetes (*n* = 4), diabetes mellitus type 2 (DM2) (*n* = 6), and six participants were managing >1 of these conditions simultaneously. The assessment of the tiered outcome for food security showed 64.3% of the sample indicated food insecurity as low security (*n* = 8) or very low security (*n* = 1) while 35.7% indicated they had high or marginal food security (*n* = 5). Participants’ mean Mental Health T-Score, mean Physical Health T-Score, and social health score are reported in Table 3 by sex, employment status, food security status, and number of chronic conditions being managed. Although financial health is described in our description of health dimensions, we did not explicitly ask participants to discuss their employment or economic status in the qualitative interview.

#### 3.2.2. Interview Theme Summary

Thematic assessment from the interviews revealed three major themes: (1) participant’s suggestion that one dimension of health is not more important, they are all interconnected, (2) health goals are centered around physical health and compliance to chronic disease management outcomes, and (3) one’s diet is most strongly tied to physical health outcomes. Additional insights connected to the participants via their quantitative survey are connected to their quotations within each theme.

#### 3.2.3. Theme 1—One Dimension of Health Is Not More Important, They Are All Interconnected

A theme of intersectionality between health dimensions was evident as 86% of participants indicated that the health dimensions discussed in the interview (social, physical, spiritual, and mental/emotional) are interrelated and important. Figure 2 shows a visual interpretation of the intersectionality of health dimensions and how they are connected as mentioned by participants. Most participants chose not to rank the dimensions of health and instead explained how the health dimensions work together or overlap. Participant 3, who rated their overall health below the average of the sample (3 out of 5) and had one of the lowest Mental Health T-Scores (38.8 ± 3.6), noted, “I’ll say this: they’re related, physical and mental health, because, well, if you don’t have one you don’t have the other.” For this participant, their mental health score is lower and they note mental and physical health are related. In this case, we know this patient is managing chronic disease. Moreover, Participant 14, who was the oldest participant of the group and had a Physical Health T-Score below average (47.7 ± 4.4), a higher than the average Mental Health T-Score (48.3 ± 3.7), and a higher financial barrier score (2 out of 3), commented,

“They are all important. Physical health is important so you can keep on living, and mental [health] not as much because I’m good, there is no worry there. Spirituality is important for me, because I am getting old, so sometimes one thinks about death being closer and one begins to be more spiritual. Not as much social [health], because I don’t have much social interaction; in a family group, we are more interacting with each other, and not strangers.”

Participant 10, who had the highest overall health rating (5/5) in the sample, also recognized intersections of health:

“I think spiritual and mental [health] go together. If your spiritual [health] is good, then it keeps the mental [health] good. You know sometimes when your mental [health] wants to come in and you want to be depressed, then you go to your spiritual [health]. You can then play music, or whatever it is to keep you positive. What you experience in all those areas shows in your physical [health].”

Similar sentiments connecting physical and mental health were provided by Participants 1, 3, 5, and 9, although these four participants all reported lower scores for Overall Health (ranging from 2–3), Mental Health T-Score (36.3 ± 3.7–41.1 ± 3.6), and Social Health (2–3) than the mean scores of the sample for the respective scores (3.1 ± 1.1, 45.2 ± 3.6, and 3.1 ± 0.6, respectively).

In contrast, other participants expressed that having one dimension of health as a priority kept other dimensions in balance, especially during major life transitions like migrating to a different country or managing their health. For example, Participant 3 recognized:

“Being in a different country where the climate changes really drastically, mental health is important for me because if you aren’t happy with yourself then you won’t have the strength to open the window or to go out and walk. Here in this country, one needs to have your mentality be the happiest it can be. Logically, life isn’t always exciting but it depends on oneself. That’s why I say mental health is the most important.”

Additionally, participants expressed that ignoring one or more health dimensions can negatively impact overall wellness, with many indicating that if one aspect of their wellness was neglected, other aspects of their health would decline. Participant 2, who reported the highest scores for overall health (5 out of 5), Physical Health T-Score (61.9 ± 5.2), and Mental Health T-Score (45.8 ± 3.6), stated, “With mental health if you are not fine, even though you might be physically well you can’t always do the right thing. I think both need to be in balance.” Concurrent with Participant 2, Participant 3 expressed:

“I think that spiritual [health is most important], or mental [health] too. I think all three. Mentally, if I’m not good, then I won’t be good with my health. Spiritually, I’m Catholic, so I say that if one isn’t good with God, then you won’t be good. All three [health dimensions] are important for me.”

As previous quotes have demonstrated, participants also noted various health dimensions linking to spiritual health. While none of the participants explicitly rated this dimension the highest, it was frequently noted. Participant 5 connected spiritual health to overall health when expressing, “… if one isn’t good with God, then you won’t be good” and another participant commenting, “I would say that the emotional state is what most affects my health … if I’m good mentally, I can help my spiritual and physical well being.”—Participant 9

When discussing health goals, Participant 3 also described the cascade of connections between dimensions of health, “… losing weight I would feel better physically, and of course mental [health], and spiritual [health] will improve, be better; what I eat affects 100% everything, my mental health and my spiritual health.”

#### 3.2.4. Theme 2—Health Goals Are Centered around Physical Health and Compliance to Chronic Disease Management Outcomes

Participants were asked about their health goals which were not connected with a health dimensions category. Of the fourteen participants, four noted goals related to losing weight, six mentioned increasing physical activity to increase their health status, and six wanted to improve chronic disease management strategies (increased blood glucose monitoring, diet, and/or blood pressure monitoring). Of the four who noted goals related to weight loss, their overall health rating was below the average of the sample (ranging from 2–3), and two of those participants had the sample’s lowest Physical Health T-Scores (29.6 ± 4.2 and 37.4 ± 4.1). Like Participant 8, some participants had more complex goals that combined physical activity, weight loss, and chronic disease management: “The goals I have right now are to lower my cholesterol level, lose weight, and to do exercise as well.” The participants with more complex goals were all women, and all rated their perceived Overall Health as a 2 out of 5.

Participants 4, 7, and 13 mentioned physical health is the most important dimension to prevent sickness or continue “feeling good”, which was achieved through diabetes management. While Participant 4 rated their physical health below the sample average, Participants 7 and 13 reported two of the higher physical health scores in the sample. Moreover, Participant 6 strongly expressed their goal to manage their chronic condition as a way to support their family: “I always have [my chronic disease] controlled so that in the future I don’t have a more chronic illness that can affect my family, because that emotionally would also affect my children who are small. I feel like they still need me.”

While most of the goals focused on physical health, the goal for Participant 9 was to find more stable employment and improve their mental and financial health. Additionally, Participant 2 mentioned they try to set goals for general health but did not provide specifics. Participant 2 had the highest overall health rating (5 out of 5), Physical Health T-Score (61.9 ± 5.2), and a lower financial barrier score (1 out of 3) than the rest of the sample. Moreover, multiple participants noted barriers to achieving their goals without being prompted. For example, Participant 3 indicated that self-discipline was a barrier to losing weight: “I need some discipline in my eating. Sometimes I lose [discipline].” Participant 1 also noted time as a barrier to being compliant to their dietary recommendations for diabetes, saying,

“Sometimes one needs to go somewhere else, and there is no time to go home, and what one does is eat out, eating in the car, so you can be able to get somewhere else. When you are hungry and you are in a rush, you eat whatever you have available and you keep going.”

Participant 4 also mentioned that location, in our case Minnesota, weather, and the gym access costs were barriers to achieving their goal of exercising more frequently: “Physical activity is important to focus on, especially when you can’t even go out and walk because of the snow. I would like more time and money to be able to pay for the gym.” This participant also had the highest financial barrier score (3 out of 3) and the lowest overall health rating (2 out of 5).

#### 3.2.5. Theme 3—One’s Diet Is Most Strongly Tied to Physical Health Outcomes

Participants noted a direct connection between food choices and overall health. Similar to health goals, when asked about their perception of diet and health outcomes, they were asked, specifically, “Do you believe that what you eat will affect your health?” Sixty-four percent (*n* = 9) of participants noted a direct connection to physical health. Three participants also indicated that food affects overall health, three participants stated food affects their mental health, and one person stated they believed food was not related to health. Participants had a range of perspectives revolving around diet and food choices. Some participants commented that food choice impacts one or more health dimensions, while others commented food impacts one dimension in a specific way.

Participant 10 noted, “I believe [what you eat will affect your health]. Some people are just healed by food.” At the same time, Participant 14 emphasized the importance of various foods in overall health: “Variety is important. When there is more variety, there is better health.” Participant 9 demonstrates that foods can induce a cascade of effects:

“If I eat something that is heavy in my body and it makes my stomach upset, it affects me emotionally, and I have my emotional [health] affected, then the spiritual [health] and mental [health] and everything else is affected.”

Most participants described food as a tool to manage chronic health conditions. Participant 7 demonstrated knowledge of food choices and its physical impact on disease management, “I feel good eating [fruits and vegetables]; what I mostly think is that my sugar levels don’t rise, that’s what I try to take care of. If I don’t pay attention to that, my sugar and blood pressure go up.” This participant was managing three chronic diseases (hypertension, dyslipidemia, and pre-diabetes) and reported a high overall health rating (5 out of 5), the highest Mental Health T-Score (53.3 ± 3.7), and a higher than average Physical Health T-Score (57.7 ± 4.9). Similarly, Participant 12, who is managing hypertension, indicates dietary interventions can reduce chronic disease development. However, continuous negative thoughts or unbalanced mental health can also affect chronic disease development: “I can’t eat many tortillas and much rice, and it’s what I like the most. But I try to not eat it. I don’t want to be diabetic because I have seen a lot of cases of diabetes. If I think over and over about [diabetes], then I know that I will get sick.” This quote also indicates the health information this participant has received in relation to dietary interventions for reducing the risk of chronic disease development.

Aside from the physical connections to food, two participants also related mental, social, and financial health as other health dimensions connected to the foods they consume. Participant 11 notes that vegetables provide positive physical and psychological energy: “In my case, since I have diabetes, the most important food is vegetables. If I eat junk food without vegetables then I feel tired. I am fatigued and sleepy, and in a bad mood in general.” Participant 6 mentions the complexities of having a large family and providing food that is beneficial for their physical and financial health:

“Yes [there is a connection between food and health], most of all physical [health] because I need to have a diet for my diabetes. Sometimes there isn’t that much money to say, ‘for my daughter, I’m going make this food’ and ‘for my son, this other one,’ because we are a big family.”

This participant had a financial barrier score of 2 out of 3, was experiencing food insecurity, and noted money as a barrier to accessing a healthful variety of foods. Conversely, the Mental Health T-Score for this participant was above the sample average (48.3 ± 3.7).

## 4. Discussion

### 4.1. General Findings and Practical Contributions

This research voices the perspectives of health and wellness for migrant Hispanic/*Latine* populations and calls for global adoption of understanding cultural consideration of health dimensions in healthcare. This study demonstrates the utilization of community-based participatory research methods to understand the perspectives of health dimensions and chronic disease treatment. Bringing to light the cultural traditions of historically voiceless communities through semi-structured ethnographic interviews represents an innovative approach to social justice in healthcare. It is imperative to recognize the paradox between the length of time living in the US and health inequities in migrant populations, inclusive of healthcare access and quality. The outcomes of this work can provide additional information for providers seeking to deliver cross-cultural or culturally competent care.

Understanding how migration affects different health dimensions can help providers with their approach to treating immigrant patients and allows for further research understanding the intersectionalities of the health dimensions. Analyzing the intersectionalities of health dimensions plays a crucial role in addressing societal challenges within the systemic framework of healthcare and research by aiming to address individuals holistically, free from bias. For providers working with Hispanic/Latine immigrant populations, screening for patient perspectives on overall, physical, mental, financial, and/or social health and inquiring about perceived quality of life could strengthen the provider’s understanding of the patient’s current state of wellness. Understanding a patient’s health from a holistic standpoint can help effectively support wellbeing and reduce medical mistrust as we seek to learn more about providing culturally inclusive, patient-centric healthcare.

The results from the qualitative interviews in this study highlight individual beliefs that multiple dimensions of health are interconnected. Of note, participants emphasized the connection between mental health and physical health. While mental health was not ranked as the highest priority, it was, however, recognized as an important piece of physical, financial, and spiritual health. Other research shows that Hispanic/*Latine* immigrant populations have a lower mental health service utilization rate compared to non-Hispanic individuals, suggesting this rate is low due to limited culturally sensitive approaches to therapeutic care, fear of microaggressions in therapy, and fear of potential deportation [2]. Our results parallel their findings that the Hispanic/*Latine* population maintains their mental health by strengthening or focusing on other health dimensions [2]. A holistic view of health, where mind, body, and spirit are central, is common for Hispanic/*Latine* populations [20,21,22,23]. For patients like Participant 3, who noted, “I’ll say this: they’re related, physical and mental health, because, well, if you don’t have one you don’t have the other,” and had one of the lowest Mental Health T-Scores, their quote suggests that mental and physical health are connected. Since they have a low Mental Health T-Score while managing a chronic disease, providers should seek to understand the status of additional health dimensions. In doing so, the patient–provider relationship strengthens, and the provider could make additional health recommendations centering stronger dimensions of health. If this participant’s social health is strong, a provider may suggest asking their friends or family for support during their diet changes, to accompany them for a walk, or encouraging a day at the gym together.

This study provides us with a small introduction to the adaptability of Hispanic/*Latine* immigrants. Their adaptability is evident in participant responses that demonstrated a shift of health dimensions to align with current priorities. For example, Participant 10 noted they change their focus to spiritual health in attempts to adapt to their current situation and balance their mental health, “… when your mental [health] wants to come in and you want to be depressed, then you go to your spiritual [health].” Religion, in this sense, is a dynamic intervention and is seen as a reservoir for strength. Many participants stated that by maintaining their connection to God, they are able to maintain other dimensions of their health.

However, adapting to the dominant culture in the US can lead to unfavorable outcomes for Hispanic/*Latine* immigrants, demonstrating the dynamic health curve immigrants encounter [3,4,5]. Multiple strategies are used to cope with lifestyle changes post-immigration. For example, a participant noted maintaining strong mental health as a strategy managing health dimensions post-immigration, “Being in a different country where the climate changes really drastically, mental health is important for me because if you aren’t happy with yourself then you won’t have the strength to open the window or to go out and walk. Here in this country, one needs to have your mentality be the happiest it can be.” Rather than ignoring their emotional distress, this participant found alternative ways to stay positive in a new country. Our results show the importance of spiritual health for some participants, noting spiritual health is essential for their overall health. These findings align with other research, stating spirituality as a coping mechanism for mental distress in Hispanic/*Latine* immigrants [22]. For healthcare providers, it is important to recognize a strong connection exists between religion and health for the Hispanic/*Latine* community, even though in the US, views on religion are vast and varied. Providers seeking to understand communities with medical mistrust should ask how they can integrate their spiritual health into their overall care.

The quantitative results of this study demonstrate the resilience of the Hispanic/*Latine* population as their Mental Health T-Scores and Physical Health T-Scores were only slightly lower than the general US population [19]. Other studies assessing perceived health in a Spanish population found results that parallel our findings; the majority of individuals surveyed with chronic health conditions perceived their health status as “good” or “very good” [24]. In the results from our subsample of participants that participated in the interview, our findings show mean Mental Health T-Scores were lower when the number of chronic diseases the participant was managing increased. However, there was no discernable difference in mean perceived overall health scores, mean Physical Health T-Scores, or mean social health scores. Participants also expressed attitudes of resilience by showing up to the interviews and being vulnerable about their experiences. These outcomes suggest that resilience and adaptability are present in a population managing chronic disease that may face higher adversity-related barriers to healthcare and food access and culturally sensitive care and/or health literacy challenges.

Beyond the social status of being an immigrant, migrants face internalized complexities surrounding their migration. These complexities often relate to the systemic barriers associated with financial health; barriers to obtaining adequate healthcare, adequate paying jobs, and food affordability [1] can affect health outcomes. One of the financial barriers this community faces is being un- or underinsured. Without insurance, patients miss preventative care appointments or have to pay high fees for emergency care, creating an ideal environment for chronic disease and increasing financial barriers/stressors. This community’s medical needs are often left unmet due to the lack of insurance. For example, a recent study found a correlation between Hispanic/*Latine* individuals without insurance and prevalence of chronic obstructive pulmonary disease [25]. Moreover, an added layer of food insecurity may exacerbate other financial barriers. As Participant 6 expressed challenges with obtaining enough food for their family, many other participants recognized food as a tool to manage chronic health conditions. The dichotomy between the need for access to and availability of culturally appropriate foods and the systemic barriers to obtaining those foods is striking. Our findings demonstrate the challenges an immigrant Hispanic/*Latine* population faces; they adapt to financial barriers and rely on other dimensions of health, resources from their country, and the strength of their cultural connection.

There can be many barriers to achieving health goals for the individual in the Hispanic/*Latine* community. A study by Amesty found that immigration, poverty, and place of residency played a significant factor in barriers to engaging in physical activity [26]. Our quantitative data show that those experiencing low/very low food security also have lower overall health perceptions, Mental Health T-Scores, and Physical Health T-Scores. Moreover, other research aligns with our results where participants noted how environmental changes like immigrating impact their health goals. For example, other research noted that men identified barriers to a healthy lifestyle including long work hours/working multiple jobs and living environments with low access to space for physical activity, while associating their weight with diet and physical activity [27]. The males in our study found compliance to an exercise regimen to be the most significant barrier despite employment and/or living conditions. Our results show that participants set goals aligned with chronic disease management. The methods to achieve their health goals are weight loss, increased exercise, or increased control of health metrics.

More specifically, 57% of participants expressed goals focused on exercise and weight loss in the qualitative interviews. Similar to our results, a qualitative study by Agne et al. found Latinas who immigrated to the US expressed interest in weight loss for both aesthetic and health-related reasons [28]. While Hispanic/Latine individuals face a disproportionate burden in rates of obesity [29], research often underscores the stigma associated with weight in diverse population groups, relying on norms centered in whiteness. Physical health is often reflected in weight status and easily becomes the central focus in primary care, especially for those managing other risk factors for chronic diseases. While excess adiposity is a risk factor in chronic disease development, our findings suggest that there may be value in shifting the focus of care to managing dimensions beyond physical health. For example, Latina adults affected by obesity rely heavily on their family, community, and healthcare system to support their goals of weight loss [30], showing the importance of social health.

An individual’s degree of acculturation to a Westernized diet is often not understood, although relevant to their overall health. This study contributes to the limited evidence connecting perceived health status and diet quality. In addition to the physical changes of moving to a new country, external factors, such as dietary choices, can contribute to chronic disease development. A national survey study found that high acculturation, or high levels of adapting to the US culture, was associated with worse diet quality [31]. With 92.9% of our participants connecting some dimension of health to the foods they eat, our findings demonstrate a relationship between diet quality and perceived health. These findings are reiterated by another study showing a significant association between perceived health status and healthy eating criteria [24]. Contrasting with our results, Alcivar et al. assessed perceived general health and diet quality of Hispanic/*Latine* individuals in Florida via the Grocery Purchase Quality Index-2016, finding that those with higher diet quality scores showed no significant difference in perceived health outcomes than those with lower diet quality scores [32]. One of our study’s participants expressed, “…I can’t eat many tortillas and much rice, and it’s what I like the most, so I try to not eat it. I don’t want to be diabetic…”, which is a common sentiment amongst our participants and in other literature [29] showing the connection between foods consumed and physical health. While this component of the study was a qualitative assessment of the association between perceived health and diet quality, our results indicate that our participants understand that food may affect health status and there is strength in having access to culturally appropriate foods for multiple dimensions of health.

### 4.2. Theoretical Implications

Recognition of and respect for cultural medicine and food practices, which may be long-standing traditions, provide a robust foundation for inclusive medical treatment for Hispanic/*Latine* patients in the US. Additionally, this study calls for future research regarding culturally appropriate approaches that decentralize Western medicine as the only option for adequate care by shifting the focus to treating all health dimensions. Having the patient teach providers how they view their individual health is vital in managing complicated and chronic diseases. Taking time to recognize facilitators and barriers to health (socially, physically, financially, spiritually, and mentally) can help support health equity for the patient. As we seek to learn more about providing inclusive healthcare via patient-centric approaches, empowering the patient to have control over their treatment options supports an understanding of the intersectionalities of health dimensions to improve patient empowerment and clinical outcomes.

### 4.3. Future Research Directions, Strengths, and Limitations

Future research could lead to the development of strong, validated tools for collecting data from community populations and providing structure for culturally tailored healthcare interventions and evidence-based healthcare to mitigate health disparities for migrant populations. Lastly, this study is able to quantify almost all health dimensions, but spirituality is difficult to assess. For the Hispanic/*Latine* population, faith is vital. More research needs to be conducted to support spiritually integrated clinical research and ways to assess the complicated and multidimensional measurement of spiritual health.

With a growing body of evidence regarding Hispanic/*Latine* immigrants, we recognize the unique experiences of Hispanic/*Latine* subgroups related to immigration and acculturation that can impact their health. However, due to the small sample size within subgroups, a limitation to the work is that we did not disaggregate the data. Moreover, there is a limited understanding of health dimensions (we did not define the health dimensions to participants in the interviews) and we recognize that spiritual health was one of the dimensions that was analyzed with the quantitative data. It was, however, addressed in the qualitative interviews and conclusions were drawn. This study’s strengths are that it is one of the first to assess multiple health dimensions from a mixed methods perspective. Moreover, our work demonstrates that community-based participatory research methods can be an applicable lens to bridge the gap regarding culturally appropriate methodologies for holistic healthcare treatments, nutrition, and clinical research practices for immigrant populations. Additionally, we offered in-person or virtual interviews to facilitate participants’ schedules and overcome barriers to transportation.

## 5. Conclusions

This research is intended to increase evidence in the literature around perceptions of wellness in Hispanic/*Latine* populations, promoting holistic healthcare for chronic disease management. Using cultural understanding to address patient participation and overall wellness can benefit individuals and populations self-identifying as Hispanic/*Latine* who migrate to the US. Healthcare professionals that provide care to diverse populations can build on this research to address individuals from a holistic perspective or use this research to inform their care practices for Hispanic/*Latine* immigrants. It is imperative to recognize social, systemic, and structural barriers that influence health outcomes for immigrants. Moreover, understanding perceptions of health for individuals that are Hispanic/Latine immigrants provides a basis for addressing medical mistrust and utilization rates for medical services. These interviews have highlighted community resilience, demonstrating that individuals can adapt to major life transitions while maintaining balance across dimensions of health. This knowledge could be implemented by actively listening to patient concerns regarding their health dimensions to improve individualized and patient-centric care promoting more equitable health outcomes for Hispanic/Latine individuals managing chronic diseases.

## Figures and Tables

**Figure 1 healthcare-12-01519-f001:**
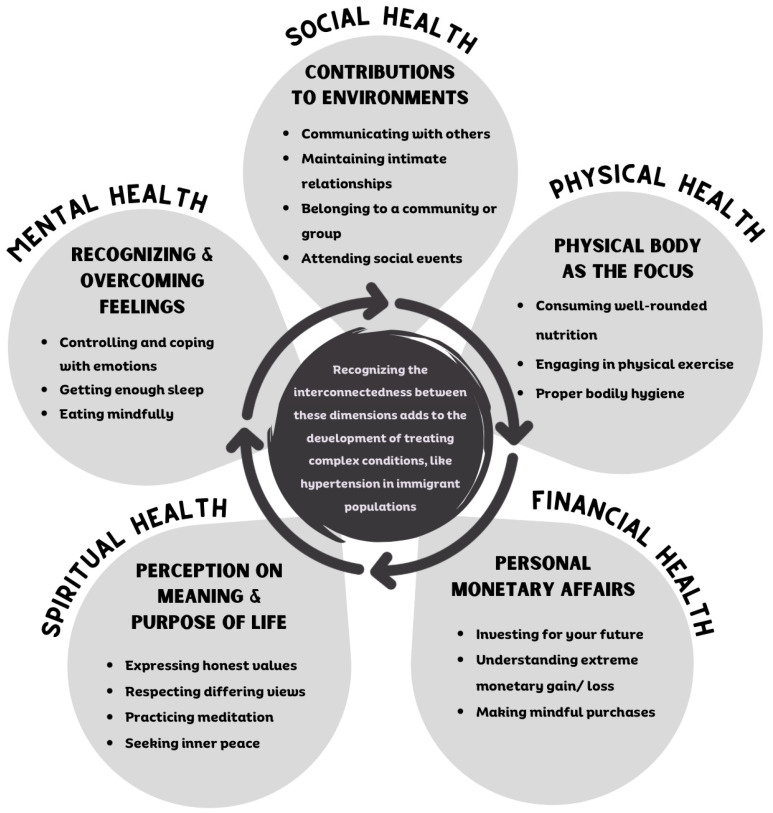
A brief overview of the five dimensions of wellness considered in this research: social health, physical health, financial health, spiritual health, and mental health [6]. Although wellness has more than five dimensions, these dimensions were used to show how health is multidimensional.

**Figure 2 healthcare-12-01519-f002:**
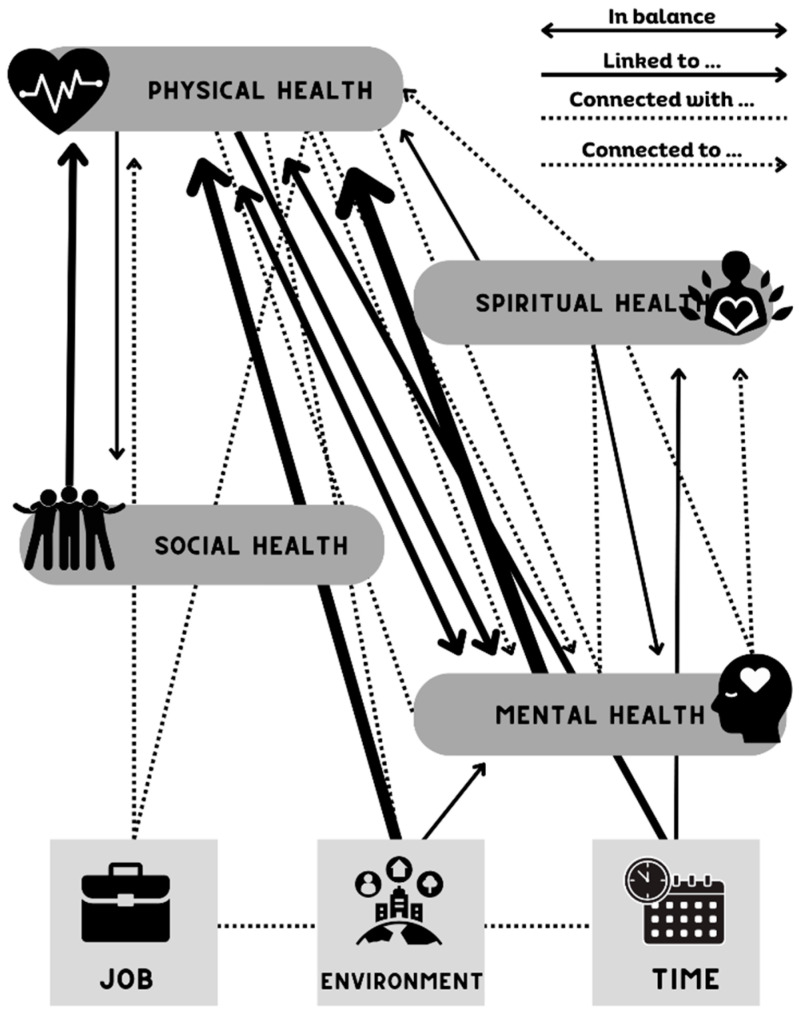
This figure provides a visual representation of how participants perceived the intersectionality of health dimensions, how the dimensions are linked (similar), connected (separate and can be influenced by the indicated dimension), and which ones are in balance. One’s job, environment, and time were aspects participants reported having an impact on the health dimensions.

**Table 1 healthcare-12-01519-t001:** Quantitative Survey Participant Demographics (*n* = 70).

Variable	N Per Category (% Sample)	Overall Health Rating (SD)	Mean Mental Health T-Score (SE)	Mean Physical Health T-Score (SE)	Social Health Rating (SD)
Overall	70	3.0 (0.9)	48.0 (3.7)	46.6 (4.5)	3.5 (0.8)
Sex					
Female	50 (71.4)	3.1 (0.9)	48.1 (3.7)	47.0 (4.5)	3.5 (0.8)
Male	20 (28.6)	3.0 (1.0)	47.6 (3.7)	45.9 (4.4)	3.5 (0.8)
Employment status					
Employed	46 (65.7)	3.1 (0.9)	48.4 (3.7)	46.2 (4.5)	3.5 (0.9)
Unemployed/retired	24 (34.3)	2.9 (0.9)	47.2 (3.7)	47.5 (4.5)	3.4 (0.5)
Food security level					
High or marginal security	36 (52.9)	3.3 (0.8) *	49.6 (3.7) *	49.0 (4.5) *	3.5 (0.6)
Low/very low security	32 (47.1)	2.8 (0.9)	46.1 (3.7)	43.9 (4.4)	3.4 (0.9)

* Indicates a significant difference between subgroups at *p* < 0.05.

**Table 2 healthcare-12-01519-t002:** Factors determining financial barrier score based on measures collected in the study’s quantitative survey and qualitative interviews to better understand financial health.

Participant (P) Number	Factor 1: Food Insecurity	Factor 2: Unemployed/Retired	Factor 3: Reported Financial Barrier	Financial Barrier Score
P1	x			1
P2			x	1
P3			x	1
P4	x	x	x	3
P5				0
P6	x		x	2
P7	x	x	x	3
P8	x	x	x	3
P9	x	x	x	3
P10	x	x	x	3
P11				0
P12		x		1
P13	x	x		2
P14	x	x		2
Participants experiencing barrier (%)	9 (64.3)	8 (57.1)	8 (57.1)	Average Score: 1.8

**Table 3 healthcare-12-01519-t003:** Demographics and chronic conditions by perceived health status for qualitative interview participants.

Variable	N Per Category (% Sample)	Overall Health Rating (SD)	Mean Mental Health T-Score (SE)	Mean Physical Health T-Score (SE)	Social Health Rating (SD)
Overall	14	3.1 (1.1)	45.2 (3.6)	46.1 (4.4)	3.1 (0.6)
Sex					
Female	11 (78.6)	2.9 (1.1)	44.8 (3.6)	44.8 (4.4)	3.1 (0.6)
Male	3 (21.4)	3.6 (1.2)	46.8 (3.7)	51.0 (4.6)	3.2 (0.8)
Employment Status					
Employed	8 (57.1)	2.9 (1.0)	43.5 (3.6)	44.5 (4.4)	3.0 (0.7)
Unemployed/retired	6 (42.9)	3.3 (1.4)	47.6 (3.7)	48.2 (4.5)	3.3 (0.5)
Food Security Level					
High or marginal security	5 (35.7)	3.0 (1.2)	43.0 (3.6)	45.8 (4.5)	2.9 (0.7)
Low/very low security	9 (64.3)	3.1 (1.2)	46.5 (3.6)	46.3 (4.4)	3.3 (0.5)
Chronic Conditions					
Managing 1	8 (57.1)	3.0 (1.2)	46.4 (3.6)	46.7 (4.4)	3.3 (0.4)
Managing 2	4 (28.6)	2.7 (0.6)	41.1 (3.7)	41.7 (4.3)	2.5 (0.5)
Managing >2	2 (14.3)	4.0 (1.4)	46.1 (3.7)	50.0 (4.6)	3.3 (1.1)

## Data Availability

The data presented in this study are available upon request from the corresponding authors. The data are not publicly available due to sensitive information regarding the documentation status of immigrants.

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
