# Peer review of "Strengthening the Voices of Hispanic/Latine Immigrants Managing Chronic Disease: A Mixed Methods Approach to Understanding Perspectives of Health"

_healthcare, 2024, doi:10.3390/healthcare12151519_

Round 1

Reviewer 1 Report

Comments and Suggestions for Authors

It is a very good study and well written and captured many areas very well. Recommendations are valid for migrant population.

Comments:

1. Inclusion criteria include patients/providers both. I was not able to capture any providers comments in this manuscript and how relevance to include providers to achieve the objectives of the study

2. Selection of study subjects was not provided in the methods section better to include it to see any biases.

3. Sample size calculation is not presented, Better to add or give a explanation on this sample size.

4. Usually migrant practice their own country diet even after migration. Is there any practices among study participants.

5. In USA, medical insurance matters a lot. Chronic disease management  depend on the availability of insurance. did you fine anything relationship in this study? Need to add few lines even in the discussion or limitations.

Author Response

Thank you for your suggestions for improvement. The authors have discussed all of your feedback at length and agree that your suggestions have allowed us to strengthen our paper. Our edits are most clear in the Word copy showing tracked changes while a clean copy is uploaded as a PDF. -AC

Comment 1: Inclusion criteria include patients/providers both. I was not able to capture any providers comments in this manuscript and how relevance to include providers to achieve the objectives of the study

Response 1: None of the providers were selected for the one-on-one interview. They did participate in the quantitative survey. All of the providers included were also Hispanic/Latine who are immigrants in the US, having similar experiences to the patients. We have added more details in the methods and results to ensure that this is clear. Please see lines 109-115 and line 172.   

Comment 2: Selection of study subjects was not provided in the methods section better to include it to see any biases.

Response 2: A more detailed explanation of the recruitment process for the quantitative survey was included in lines 119-121; a more detailed recruitment process for the qualitative interviews was included in lines 178-189. 

Comment 3: Sample size calculation is not presented, Better to add or give a explanation on this sample size.

Response 3: A sentence has been added for the quantitative sample size (see lines 122-124) and for qualitative interviews (line 212). 

Comment 4: Usually migrant practice their own country diet even after migration. Is there any practices among study participants.

Response 4: We have added more context in the introduction, “Additionally, family and social networks provide healthcare support. For example, Hispanic/Latine individuals who migrate to the US are often supported by their community to follow their traditional dietary patterns and maintain healthful cultural practices.” in lines 77-80. 

Comment 5: In USA, medical insurance matters a lot. Chronic disease management  depend on the availability of insurance. did you fine anything relationship in this study? Need to add few lines even in the discussion or limitations.

Response 5: Yes, insurance is a very important piece of healthcare delivery in the US. We added lines to the methods to better understand the participant demographics as all of the patients of the community health clinic are uninsured (lines 111-115). We also added a paragraph in the discussion to address this: 

Beyond the social status of being an immigrant, migrants face internalized complexities surrounding their migration. These complexities often relate to the systemic barriers related to financial health;  barriers to obtaining adequate healthcare, adequate paying jobs, and food affordability [1] can affect health outcomes. One of the financial barriers this community faces is being un- or under-insured. Without insurance, patients miss preventative care appointments or have to pay high fees for emergency care, creating an ideal environment for chronic disease and increasing financial barriers/stressors. This community’s medical needs are often left unmet due to the lack of insurance. For example, a recent study found a correlation between Hispanic/Latine individuals without insurance and prevalence of chronic obstructive pulmonary disease [25]. Moreover, an added layer of food insecurity may exacerbate other financial barriers. As Participant 6 expressed challenges with obtaining enough food for their family, many other participants recognized food as a tool to manage chronic health conditions. The dichotomy between the need for access and availability of culturally-appropriate foods and the systemic barriers to obtaining those foods is striking. Our findings demonstrate the challenges an immigrant Hispanic/Latine population face; they adapt to financial barriers and rely on other dimensions of health, resources from their country, and the strength of their cultural connection.

Reviewer 2 Report

Comments and Suggestions for Authors

Interesting topic but leaves the reader wanting to know more through analysis and less description. 

The beginning/contextualizing is unclear and should be reframed for clarity.

There are several typos to check.

Integrating the references into the text would bring the author(s) ideas and voice to the fore, at the moment it is a bit lost among the use of references.

It would be good to demonstrate the authors understanding/command of the topic and literature with more authority.

The methods uncleared how design, setting, approach, and analysis.

There are some very insightful quotes from the participants that could have been used in analysis and unpicked more.

More in-depth analysis in the paper is required, some things are hard for the reader to link up as it is only stated as fact/finding but it needs to be backed up with evidence or example.

Should be added and cleared sub-heading of discussion, including general finding discussion, theoretical implications, practical contributions, and Limitations and further study.

Conclusion should be clear how provided insights into the main findings with summarization.

Comments on the Quality of English Language

Some sentences are missing meaning, errors, and wrong meaning.

Author Response

Thank you for your suggestions for improvement. The authors have discussed all of your feedback at length and agree that your suggestions have allowed us to strengthen our paper. Our edits are most clear in the Word copy showing tracked changes while a clean copy is uploaded as a PDF. -AC

Comment 1: The beginning/contextualizing is unclear and should be reframed for clarity.

Response 1: We have added more context in the introduction, “Additionally, family and social networks provide healthcare support. For example, Hispanic/Latine individuals who migrate to the US are often supported by their community to follow their traditional dietary patterns and maintain healthful cultural practices.” in lines 77-80. We also added a transition sentence in lines 44-45 to add more clarity and tie in the dietary portion of the study. 

Comment 2: There are several typos to check.

Response 2: We have thoroughly reviewed and revised the language and spelling in the manuscript and hope it is more clear. 

Comment 3: Integrating the references into the text would bring the author(s) ideas and voice to the fore, at the moment it is a bit lost among the use of references.

Response 3: We had reduced the number of times the authors of other studies have been referenced and integrated their results into the work that we have done. We are hopeful that this has brough forward the author voice, especially in the discussion. 

Comment 4: It would be good to demonstrate the authors understanding/command of the topic and literature with more authority.

Response 4: We removed author names when talking about relevant studies and instead integrated the information from the studies to support our findings. We have revised and attempted to more clearly articulate the discussion. We are hopeful that this has brought forward the author’s understanding of the literature, especially in the discussion. 

Comment 5: The methods uncleared how design, setting, approach, and analysis.

Response 5: More detail has been added to the methods for clarity in the processes in both phases.  We have labeled the Phases as Phase 1 and Phase 2 to help support the difference between the quantitative and qualitative methods. We have also expanded portions of the methods to add clarity. Please see lines 111-115, 119-124, 178-180, 198-205, and 210-212. 

Comment 6: There are some very insightful quotes from the participants that could have been used in analysis and unpicked more.

Response 6: We have unpacked quotes we felt were meaningful in the discussion to add a more robust analysis. Please see lines 496-506, 509-515, 519-527 and 561-563. 

Comment 7: More in-depth analysis in the paper is required, some things are hard for the reader to link up as it is only stated as fact/finding but it needs to be backed up with evidence or example.

Response 7: More examples have been added in the discussion to connect back to our study’s findings. Please see the examples from the previous comment (lines 496-506, 509-515, 519-527 and 561-563).

Comment 8: Should be added and cleared sub-heading of discussion, including general finding discussion, theoretical implications, practical contributions, and Limitations and further study.

Response 8: Thank you for this suggestion. We have added 3 sub-headings in the discussion (after moving some content) and hope this provides more structure. 

Comment 9: Conclusion should be clear how provided insights into the main findings with summarization.

Response 9: Thank you for this suggestion. Part of the previous conclusion was moved to the discussion and the conclusion was reconsidered for clarity and continuity. We have tied in a major point of medical mistrust to ensure there is clear alignment. Please see lines 662-671. 

Reviewer 3 Report

Comments and Suggestions for Authors

This is an original and interesting study, though some changes seem advisable.

You can find my suggestions below.

1. Affiliations: please give details (Department, Unit, City, etc.)

2. Paragraph 2.1.1, line 99: "a patient or provider"; paragraph 2.1.2, line 105 "70 patients and providers". I do not understand. Does it mean that to enter the study one needed to be either a patient or a provider? If it is so how many were patients and how many were providers out of 70?

I think that enrolling only patients would have been better, also in consideration of the conclusions the Authors draw on resilience.

3. Paragraph 2.2.1, lines 178-80 "The interviews (in person=4, virtual=10) that were coducted in Spanish,..." This phrase is not clear.

4. Table 1. Why the Authors use 2 times SD and 2 times SE?

5. Paragraphs 3.2.3, 3.2.4 and 3.2.5: the theme of obesity clearly emerges as important for many patients (see for example lines 338, 344-45, 369-70, 375-76, 386-87) and this is also underlined in the discussion (see page 13). This is intersting, also in the light of the fact that the prevalence of obesity seems to be high in Hispanic/Latine communities. It would therefore be worth to have further details on obesity in the population of this study.

5. Conclusions: in my opinion these are too long and resemble more to general considerations than short and pertinent conclusions based on the data of the sudy. Please consider rewriting this paragraph.

Author Response

Thank you for your suggestions for improvement. The authors have discussed all of your feedback at length and agree that your suggestions have allowed us to strengthen our paper. Our edits are most clear in the Word copy showing tracked changes while a clean copy is uploaded as a PDF. -AC

Comment 1: Affiliations: please give details (Department, Unit, City, etc.)

Response 1: All affiliations have been updated with more detail. 

Comment 2: Paragraph 2.1.1, line 99: "a patient or provider"; paragraph 2.1.2, line 105 "70 patients and providers". I do not understand. Does it mean that to enter the study one needed to be either a patient or a provider? If it is so how many were patients and how many were providers out of 70?I think that enrolling only patients would have been better, also in consideration of the conclusions the Authors draw on resilience.

Response 2: None of the providers were selected for the one-on-one interview. They did participate in the quantitative survey. All of the providers included were also Hispanic/Latine who are immigrants in the US, having similar experiences to the patients. We have added more details in the methods and results to ensure that this is clear. Please see lines 109-115 and line 172.   

Comment 3: Paragraph 2.2.1, lines 178-80 "The interviews (in person=4, virtual=10) that were coducted in Spanish,..." This phrase is not clear.

Response 3: This portion of the methods was expanded for clarity. Please see lines 197-207. “The interviews that were conducted in Spanish, were translated by native Spanish speakers (A.R. and C.G.-H.), and professionally transcribed prior to data analysis. Due to varying schedules, barriers to transportation, and time, all participants were given the option to complete the interview in person or virtually (in person n=4, virtual n=10). 

Comment 4: Table 1. Why the Authors use 2 times SD and 2 times SE?

Response 4: The T-scores utilized SE as part of their tool methodology. It didn't apply to our other tools since this was comparative to within study measures and the global T scores are accurate the mean of a sample is compared to the true population mean.

Comment 5: Paragraphs 3.2.3, 3.2.4 and 3.2.5: the theme of obesity clearly emerges as important for many patients (see for example lines 338, 344-45, 369-70, 375-76, 386-87) and this is also underlined in the discussion (see page 13). This is interesting, also in the light of the fact that the prevalence of obesity seems to be high in Hispanic/Latine communities. It would therefore be worth to have further details on obesity in the population of this study.

Response 5: Thank you for this comment– it truly is critical to consider and is also met with challenges for us as authors. Because physical health (which is often stigmatized via white ideals for appropriate” body weight) is often the medical priority, we wanted to reduce our focus on the topic of obesity and centralize a more holistic approach to viewing health. We have added a more clear paragraph that centers the theme of goals centered around weight loss, focusing on the risks for chronic disease management. Please see lines 584-593. 

Comment 6: Conclusions: in my opinion these are too long and resemble more to general considerations than short and pertinent conclusions based on the data of the sudy. Please consider rewriting this paragraph.

Response 6: Thank you for this suggestion. Part of the previous conclusion was moved to the discussion and the conclusion was reconsidered for clarity and continuity. Please see lines 655-671. 

Round 2

Reviewer 2 Report

Comments and Suggestions for Authors

All comments are revised and suited for publication in Healthcare. 

Comments on the Quality of English Language

The minor editing is required.